# Revealing the mechanism for covalent inhibition of glycoside hydrolases by carbasugars at an atomic level

Weiwu Ren [1], Robert Pengelly[2], Marco Farren-Dai[1], Saeideh Shamsi Kazem Abadi[3], Verena Oehler[2], Oluwafemi Akintola[1], Jason Draper[1], Michael Meanwell[1], Saswati Chakladar[1], Katarzyna Świderek [4], Vicent Moliner [4], Robert Britton[1], Tracey M. Gloster [2] & Andrew J. Bennet [1]

Mechanism-based glycoside hydrolase inhibitors are carbohydrate analogs that mimic the natural substrate's structure. Their covalent bond formation with the glycoside hydrolase makes these compounds excellent tools for chemical biology and potential drug candidates. Here we report the synthesis of cyclohexene-based α-galactopyranoside mimics and the kinetic and structural characterization of their inhibitory activity toward an α-galactosidase from *Thermotoga maritima* (*Tm*GalA). By solving the structures of several enzyme-bound species during mechanism-based covalent inhibition of *Tm*GalA, we show that the Michaelis complexes for intact inhibitor and product have half-chair ($^2H_3$) conformations for the cyclohexene fragment, while the covalently linked intermediate adopts a flattened half-chair ($^2H_3$) conformation. Hybrid QM/MM calculations confirm the structural and electronic properties of the enzyme-bound species and provide insight into key interactions in the enzyme-active site. These insights should stimulate the design of mechanism-based glycoside hydrolase inhibitors with tailored chemical properties.

[1] Department of Chemistry, Simon Fraser University, 8888 University Drive, Burnaby, BC V5A 1S6, Canada. [2] Biomedical Sciences Research Complex, University of St Andrews, North Haugh, St Andrews, Fife KY16 9ST, UK. [3] Department of Molecular Biology and Biochemistry, Simon Fraser University, 8888 University Drive, Burnaby, BC V5A 1S6, Canada. [4] Department de Química Física i Analítica, Universitat Jaume I, 12071 Castellón, Spain. Correspondence and requests for materials should be addressed to R.B. (email: rbritton@sfu.ca) or to T.M.G. (email: tmg@st-andrews.ac.uk) or to A.J.B. (email: bennet@sfu.ca)

Of the three main biological polymeric building blocks, carbohydrates are the most structurally diverse. Unlike DNA and proteins, their sequence is not templated and instead relies on the activity and specificity of enzyme-catalyzed processes that use carbohydrates and their derivatives as substrates[1]. For example, glycoside hydrolases (GHs)[2] are ubiquitous in nature and catalyze the removal of carbohydrates from a range of biomolecules. Thus GHs are not only critical for digesting carbohydrates and degrading plant biomass but are also key players in pathogen infection, antibacterial defense, and many other essential cellular processes[3]. As a result, natural product inhibitors of GHs, including castanospermine[4], mannostatin A[5], nojirimycin[6], and acarbose[7], are often pursued as therapeutics (Fig. 1a). For example, starting with the natural product Neu2en5Ac[8] medicinal chemical approaches gave rise to the influenza therapeutic oseltamivir (Fig. 1b)[9].

Of note, many GH inhibitors incorporate a basic nitrogen atom[6] that mimics the nascent charge of pyranosylium ion-like transition states[10] (Fig. 2a). Structural elements that mimic enzyme-bound substrates[11] and preclude a ground state chair conformation include sp²-hybridized ring atoms[12], five-membered rings[5], or bicyclic scaffolds[4,13] and are also common features in GH inhibitors. Considering that GHs exhibit high catalytic proficiencies (~$10^{17} M^{-1}$)[14,15], deciphering structures along the reaction pathway can lead to improved understanding of their catalytic mechanism and profoundly affect inhibitor design[16]. Of the GH mechanisms that catalyze glycosidic bond hydrolysis, most that occur with retention of stereochemistry rely on two active site aspartic acid (Asp) and/or glutamic acid (Glu) residues (Fig. 2a)[1,10,17,18]. These enzymes employ sequential $S_N2$-like reactions, each involving an inversion of configuration, where the first generates a covalent glycosyl-enzyme intermediate and the second one hydrolyzes the intermediate[1,10,19–21]. Previously, we exploited the ability of GHs to stabilize anomeric positive charge development during glycosylation and deglycosylation in the design of cyclohexene[22] (e.g., 1) and cyclopropylmethyl[22–24]

mechanism-based covalent GH inhibitors. The cyclohexene inhibitors likely undergo a pseudo-glucosylation reaction as shown for 1 (Fig. 2b)[22]. Recently, Danby and Withers reported that the related cyclohexene carbaglucose analog 2 was a substrate for several β-glucosidases (Fig. 2c)[25]. Here we present the synthesis of three cyclohexene-based mimics of galactose including a 2-deoxy-2-fluorogalactose analog and the kinetic characterization of their reactions with an α-galactosidase from *Thermotoga maritima* (*Tm*GalA). The GH inhibitory activity of these compounds is compared with that of a 2-deoxy-2-fluoroglycoside and a cyclophellitol analog, both established inactivators[26,27]. We also present the structural characterization of our carbasugars with *Tm*GalA in the form of Michaelis complexes for the inhibitor (with an active site mutant), the reaction products, and most critically, the covalent adduct formed during the turnover of our cyclohexene mimic of 2-deoxy-2-fluorogalactose. Using a combination of quantum mechanics/molecular mechanics (QM/MM) methods, we also provide support for the X-ray structural results by localizing and characterizing the relevant enzyme-bound states.

## Results

**Synthesis of cyclohexene galactose analogs.** Previously, we have reported that the cyclohexene carbasugar 1 is a covalent inhibitor of yeast α-glucosidase (GH13 family)[22] and that the reaction takes place through a covalent enzyme-bound intermediate, which undergoes slow hydrolysis to regenerate active enzyme. That is, covalent inhibitors result in reversible loss of enzyme activity, a process that is irreversible with covalent inactivators. As a result, we hypothesized that a *galacto*-configured analog would covalently label a GH36 α-galactosidase, a member of GH clan-D, that has as its main structural element a (β/α)₈ fold, which is the same protein fold as found for family GH13. Notably, these two GH families bind bicyclo[4.1.0]heptyl amines (with the appropriate stereochemistry) tightly[13,28]. Initially, we used a tetra-O-benzyl-cyclohexenol[24] that was arylated via an $S_N$Ar reaction after debenzylation with BCl₃ in CH₂Cl₂ to give covalent inhibitor 3a. Unfortunately, after performing a variety of crystal soaking experiments with inhibitor 3a, only structures of the Michaelis complex (with an active site mutant) and product complex were obtained and no covalent adduct was observed (vide infra). In an effort to characterize structurally the covalent intermediate, we elected to (i) enhance the leaving group ability of the pseudo-aglycone and (ii) structurally modify the substituents on the cyclohexene ring in an effort to increase the lifetime of the covalent intermediate. Thus we targeted the *galacto* configured 2,4-dinitrophenyl ether 3b and the corresponding 2-fluoro analog 4, with the expectation that covalent intermediates would form faster and that the 2-fluoro group in 4 would decrease the rate of hydrolysis of the covalent intermediate[29,30].

Synthesis of the 2-hydroxycarbasugar 3b started with 2-deoxy-L-ribose-derived aldehyde 9, which was subjected to a proline-catalyzed α-chlorination-aldol reaction (Fig. 3)[31,32] that effects a dynamic kinetic resolution involving the racemization of chloroaldehydes 10 and 11 and delivered the *anti*-aldol-*syn*-chlorohydrin 13 in good yield and excellent diastereoselectivity (dr > 20:1)[31]. With chlorohydrin 13 in hand, several olefination methods were examined and we found that a Julia–Kocienski reaction[31] using the lithium anion derived from methylsulfonyl phenyltetrazole delivered the diene 14 without requiring protection of the secondary alcohol function. Unfortunately, attempts to effect a ring closing metathesis (RCM)[33–35] on this material using various catalysts returned only starting material. Also, removal of the TIPS protecting group (TBAF) followed by treatment with Grubbs' first[33] or second-generation[34] catalyst resulted primarily

**Fig. 1** Select competitive inhibitors of glycoside hydrolases. **a** Representative natural products. **b** Natural product (Neu2en5Ac) inspired therapeutic agent (oseltamivir)

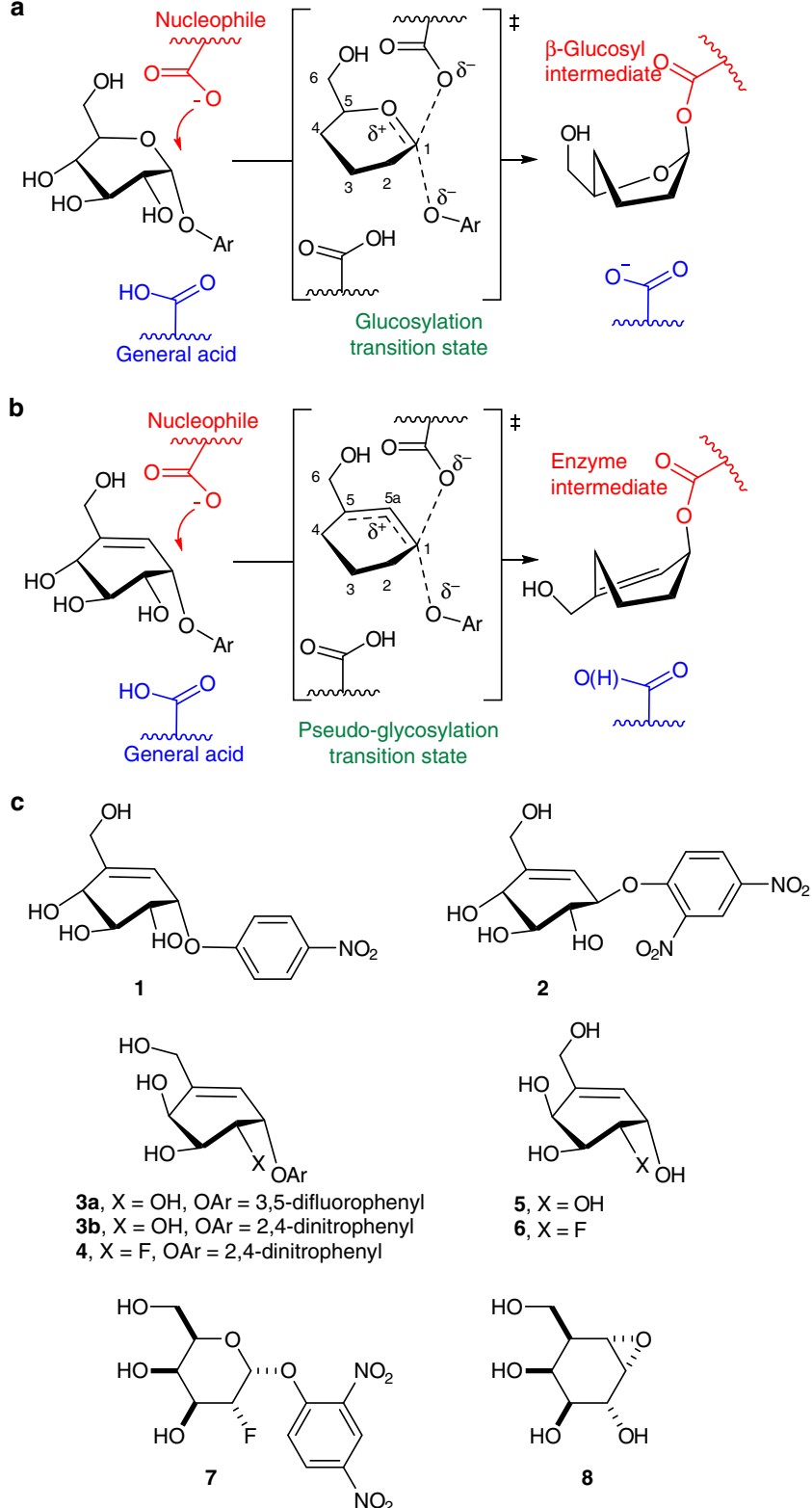

**Fig. 2** Glycoside hydrolase catalysis and chemical structures for select inhibitors. **a** Mechanism for glucosylation with a natural substrate analog. **b** Proposed mechanism of pseudo-glucosylation for the allylic α-glucoside mimic **1**. **c** Structures of mechanism-based 4-nitrophenyl covalent inhibitor **1**; 2,4-dinitrophenyl β-glucoside analog **2**; the α-galactose analog inhibitors **3a**, **b**; the 2-deoxy-2-fluoro covalent inhibitors **4**; **5**, and **6** the hydrolyzed products for the α-galactose analog inhibitors **3** and **4**, respectively; and two conventional glycoside hydrolase inactivators: 2-deoxy-2-fluorogalactoside (**7**) and cyclophellitol analog (**8**). For clarity, most hydroxyl groups are not shown for the transition state or intermediate in **a**, **b**

**Fig. 3** Synthesis of the cyclohexene carbagalactose analog **3b**

in isomerization to the ethyl ketone **17**[36,37]. In an effort to circumvent these problems and facilitate the desired RCM reaction, we converted chlorohydrin **14** into the corresponding *syn*-epoxide by treatment with CsOH in hot EtOH-H$_2$O. While the TIPS-protected epoxyalcohol was not a productive substrate for a RCM, following the removal of the silyl protecting group the corresponding alcohol **15** was readily transformed into cyclohexene **16** using either Stewart–Grubbs'[35] or Grubbs' second-generation catalyst[34]. From here, conversion into the protected carbasugar simply involved reaction of the epoxy alcohol **16** with CO$_2$ and CsCO$_3$ in warm dimethylformamide (DMF), which afforded the carbonate **18** in excellent yield. Finally, removal of the carbonate and reaction of the resulting triol with 2,4-dinitrofluorobenzene[23] afforded a mixture of arylated carbasugars from which the targeted dinitrophenyl adduct **3b** could be isolated by preparative thin-layer chromatography (Supplementary Methods and Supplementary Figs. 8–21).

The corresponding 2-fluorocarbasugar **4** was accessed following a similar route to that described above (Fig. 4). However, we modified our previously reported α-chlorination aldol reaction by first effecting a proline-catalyzed α-fluorination[32,38–40] of aldehyde **9** using Selectfluor in DMF followed directly by reaction of the resulting unstable α-fluoroaldehyde (not shown) with

dioxanone **12** in CH$_2$Cl$_2$. This subsequent proline-catalyzed aldol reaction proceeded smoothly to afford fluorohydrin **19** as a single diastereomer.

To mitigate degradation of the unstable aldol adduct **19**, the crude reaction product was subjected directly to the optimized Julia–Kocienski olefination conditions[41], which afforded the diene **20** as a stable and isolable intermediate. From here, acetylation of the free alcohol provided diene **21**, which following removal of the TIPS protecting group, underwent RCM using Grubbs' second-generation catalyst[34] to afford protected 2-fluorocarbasugar **22** in excellent overall yield. Finally, arylation of the free alcohol function and global deprotection provided the target 2-fluorocarbasugar **4** in good yield (Supplementary Methods and Supplementary Figs. 22–29).

## Evaluation of GH36 α-galactosidase covalent inhibitors.

With the target molecules in hand, we first assayed the GH36 family α-galactosidase from *T. maritima* (*Tm*GalA) with activated carbasugar analog **3b**, which was turned over within 10 min and is therefore considered a poor substrate for *Tm*GalA (Table 1, Supplementary Fig. 1). Of note, we observed (using ¹H nuclear magnetic resonance spectroscopy in D$_2$O) that compound **3b**

**Fig. 4** Synthesis of the 2-deoxy-2-fluorocarbagalactose analog **4**

| Table 1 Kinetic parameters for the *Tm*GalA-catalyzed reactions with 3b and 4 | | | |
|---|---|---|---|
| **Hydrolysis substrates** | $k_{cat}$ (s$^{-1}$) | $k_{cat}/K_m$ (M$^{-1}$ s$^{-1}$) | $K_m$ (μM) |
| **3b** | $(3.78 \pm 0.06) \times 10^{-2}$ [a] | $(2.25 \pm 0.19) \times 10^4$ | $1.68 \pm 0.14$ |
| **4** | $(2.29 \pm 0.11) \times 10^{-4}$ [b] | $56 \pm 13$ | $4.1 \pm 0.9$ |
| **7** | $3.04 \pm 0.32$ | $1200 \pm 160$ | $253 \pm 8$ |
| 4-NPG[c] (D387A) | $(3.60 \pm 0.03) \times 10^{-2}$ | $196 \pm 6$ | $184 \pm 7$ |
| **Inhibitor/inactivator** | $k_{inact}$ (s$^{-1}$) | $k_{inact}/K_i$ (M$^{-1}$ s$^{-1}$) | $K_i$ (μM) |
| **4** | id | $143 \pm 5$ | id |
| **8** | $(1.38 \pm 0.05) \times 10^{-4}$ | $0.86 \pm 0.10$ | $161 \pm 18$ |

Conditions for all experiments were $T = 37$ °C in 50 mM HEPES buffer, pH 7.4, quoted errors (±) are the standard errors for the fits of kinetic data to either the Michaelis-Menten equation (all values except for inactivation by **4**) or a linear equation
id; indeterminable
[a] $k_{cat} = (9.7 \pm 2.3) \times 10^{-2}$ s$^{-1}$ corrected value based on the inactivation data for **4**, which does not assume 100% activity
[b] $k_{cat} = (5.9 \pm 1.3) \times 10^{-4}$ s$^{-1}$ calculated from the inactivation data for **4**
[c] 4-NPG = 4-nitrophenyl α-d-galactopyranoside; values for wild-type *Tm*GalA-catalyzed hydrolysis of 4-NPG are: $k_{cat} = 33$ s$^{-1}$, and $k_{cat}/K_m = 6.2 \times 10^5$ M$^{-1}$ s$^{-1}$ (ref.[45])

undergoes an intramolecular rearrangement to give the isomer in which the aryloxy group is attached to C-2 (i.e., **23**, Supplementary Fig. 2) with a final equilibrium position (in D$_2$O) is ~2:1 in favor of **3b** (Supplementary Figs. 30 and 31). As a result, our measured kinetic parameters $k_{cat}$ and $k_{cat}/K_m$ for the hydrolysis of **3b** by *Tm*GalA (Table 1) are lower limits if, as expected, the enzyme does not bind the C2 aryloxy isomer.

Next we measured the kinetic parameters for turnover of **4** by *Tm*GalA (Table 1: Supplementary Fig. 3a shows a standard Michaelis–Menten plot for this hydrolysis reaction) and found that the apparent binding constant ($K_m$) is in the low μM range. This result suggests that pseudo-deglycosylation of **4** is rate-limiting[25] and that the replacement of the 2-hydroxyl group with a fluorine results in accumulation of the covalent intermediate. Consequently, we undertook a sequential mixing stopped-flow experiment to measure the rate constants for the pseudo-

glycosylation reaction[42]. Plots of remaining α-galactosidase activity against incubation time for four concentrations of **4** are shown in Supplementary Fig. 4. It is clear from these stopped-flow data of rate constant versus inhibitor concentration (Supplementary Fig. 3b) that we can only calculate the second-order rate constant $k_{inact}/K_i$ from these covalent labeling experiments. We next compared our covalent inhibitor with the corresponding α-galacto analogs from two well-characterized classes of GH inactivators. Specifically, we synthesized both 2,4-dinitrophenyl 2-deoxy-2-fluoro-α-D-galactopyranoside **7** and the cyclophellitol analog **8** following literature procedures[43,44] with minor modifications (Supplementary Methods and Supplementary Figs. 32 and 33). Of note, we observed no time-dependent decrease in *Tm*GalA activity in the presence of **7**; rather we determined that it is a poor substrate for GH36 α-galactosidase (Table 1, Supplementary Fig. 5). As reported for the human

GH27 enzyme[44], cyclophellitol analog **8** forms the covalent intermediate with GH36 α-galactosidase in a time-dependent manner that is characterized by a second-order rate constant that is about 23,000-fold lower than that for **3b** (Table 1, Supplementary Figs. 6 and 7).

**Structural insights into carbasugar inhibition of *Tm*GalA.** With the aim of delineating the conformational itinerary for intermediates along the reaction path, we undertook structural studies to observe enzyme-bound species during the covalent inhibition of *Tm*GalA by carbasugar **3a**. In order to obtain a complex with intact **3a**, we employed a D387A mutant of *Tm*GalA in which the acid/base aspartic acid is mutated to alanine; this mutant displays an approximate 3100-fold reduction in its second-order rate constant ($k_{cat}/K_m$) relative to wild-type[45] for the hydrolysis of 4-nitrophenyl α-D-galactopyranoside (Table 1). Crystals of *Tm*GalA D387A that were soaked with **3a** for 2 h diffracted to 2.20 Å resolution and the structure was solved using molecular replacement with apo *Tm*GalA (PDB 5M0X) as the search model. We note that the active site contains electron density consistent with a Michaelis complex of an intact molecule of **3a** bound (Fig. 5a). The carbasugar moiety displays a $^2H_3$ half

chair conformation with key interactions between the C6-OH with Asp221; the C4-OH with Trp257, Asp220, and Lys325; and the C3-OH with Lys325, Arg383, and Tyr191. The C2-OH and the 3,5-difluorophenyl leaving group of **3a** form no interactions with *Tm*GalA, and **3a** itself has no interactions with bound water molecules.

To obtain structural information regarding the enzyme-bound species, **3a** was also soaked with wild-type *Tm*GalA crystals. Following an overnight soak with **3a**, the structure solved using data to 1.22 Å resolution revealed electron density consistent with the hydrolyzed form **5** in the active site of *Tm*GalA (Fig. 5b). The interactions in the active site were essentially the same as for **3a** (Fig. 5c), except for the addition of hydrogen bonds between the C2-OH with Arg383 and Asp387, and C1-OH with Asp387 and a water molecule. Thus, within the overnight timeframe **3a** had been hydrolyzed with retention of stereochemistry at the pseudo-anomeric center, yielding a Michaelis complex of the reaction product. In an effort to observe the covalent intermediate derived from **3a**, crystals were soaked for various time periods (from 5 s to 8 h) with **3a** but did not yield any additional information. We therefore turned our attention to obtaining structural insights of *Tm*GalA with **4**, for which the covalent intermediate has a longer half-life (vide infra). Co-crystallization of the D387A mutant with

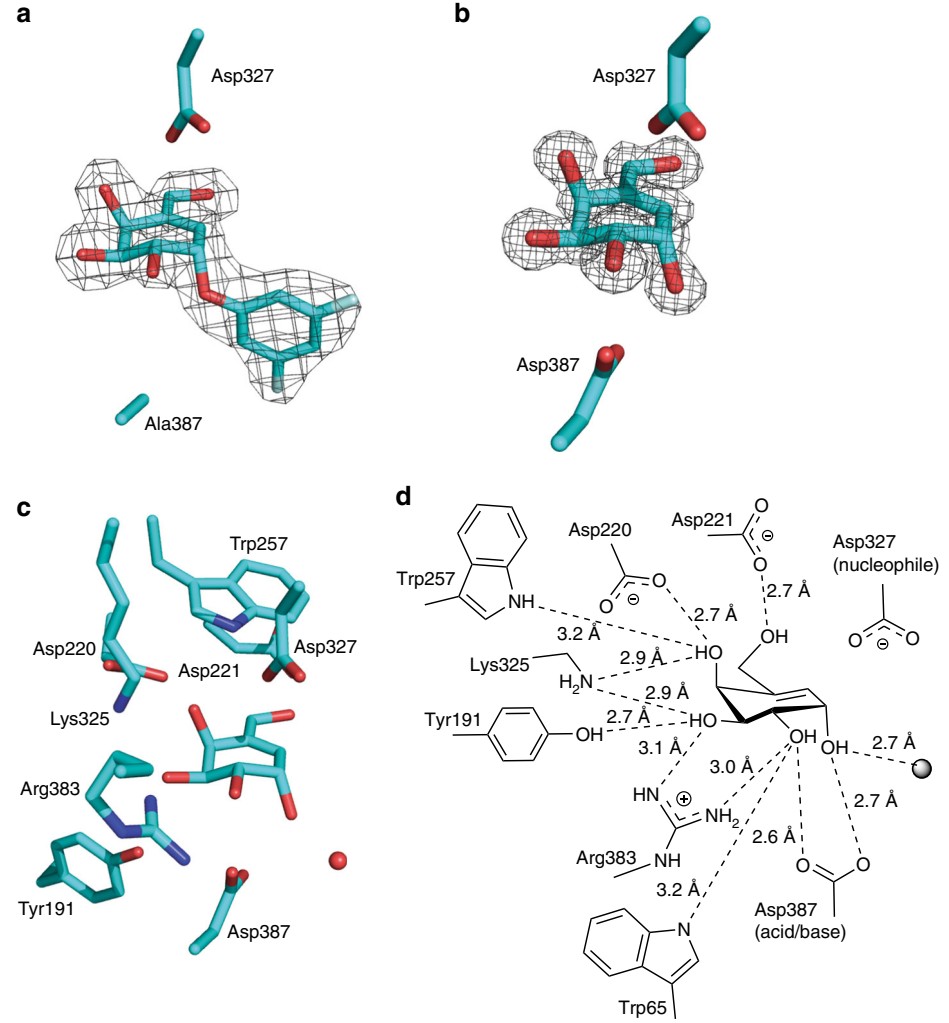

**Fig. 5** Structure of *Tm*GalA in complex with **3a** and **5**. **a** Structure of *Tm*GalA mutant D387A in complex with intact **3a**. **b** Structure of *Tm*GalA in complex with hydrolyzed inhibitor **5**. The maximum likelihood/$\sigma_A$ weighted $2F_{obs}-F_{calc}$ electron density map is contoured at 1.2 sigma in **a** and 2 sigma in **b**. The catalytic residues D327 and D387 (or D387A) residues are shown. **c**, **d** Structure of *Tm*GalA in complex with hydrolyzed inhibitor **5** illustrating active site residues that hydrogen bond with the inhibitor. The red sphere in **c** and gray sphere in **d** represent a water molecule

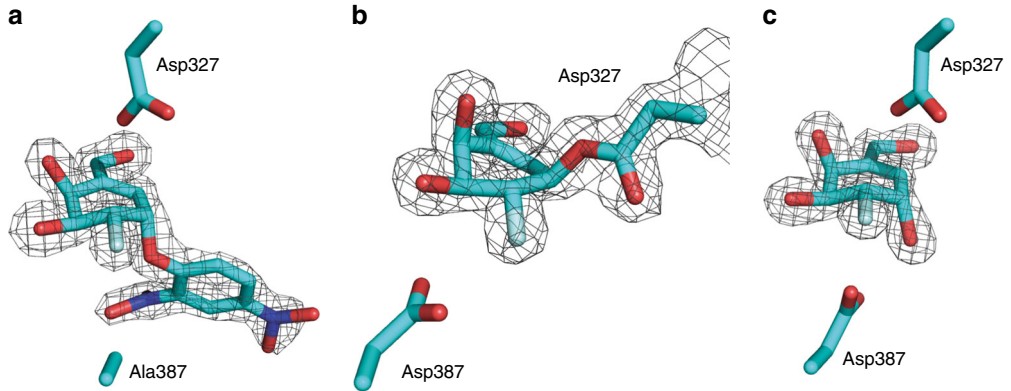

**Fig. 6** Structure of *Tm*GalA in complex with **4**. **a** Structure of *Tm*GalA mutant D387A in complex with intact **4**. **b** Structure of *Tm*GalA in complex with 2-deoxy-2-fluorocarbagalactose fragment of **4** covalently bound to the nucleophile D327. **c** Structure of *Tm*GalA in complex with hydrolyzed inhibitor **6**. The maximum likelihood/$\sigma_A$ weighted $2F_{obs}-F_{calc}$ electron density map is contoured at 1.5 sigma in all cases. The catalytic residues D327 and D387 (or D387A) residues are shown

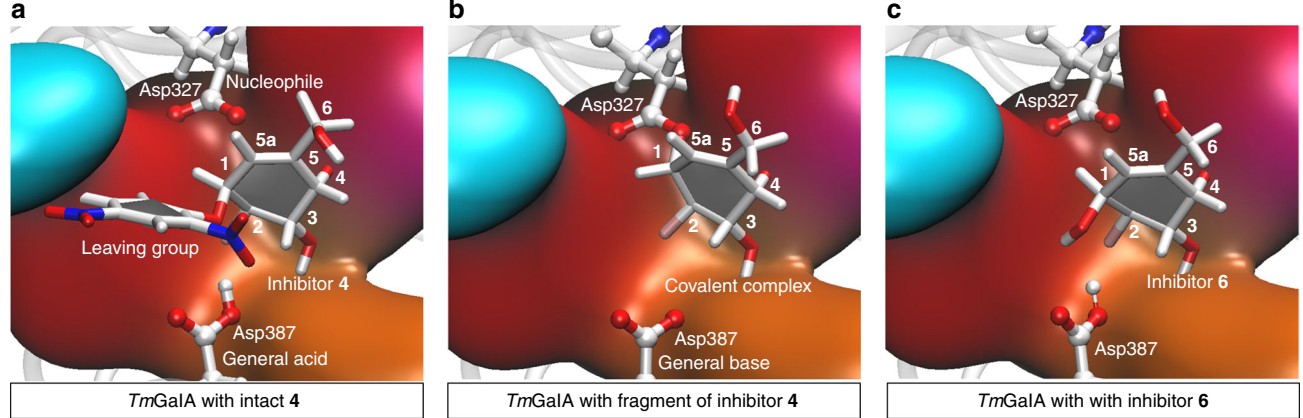

**Fig. 7** Structural snapshots of the active site of *Tm*GalA. The enzyme's active site is shown in complex with: **a** **4** (or **E:I**); **b** 2-deoxy-2-fluorocarbagalactose fragment of **4** covalently bound to the nucleophile Asp327 (or **E-I**) and **c** hydrolyzed inhibitor **6** (or **E:P**). Structures were optimized at DFT/MM level of theory

**4** produced crystals that diffracted to 1.72 Å resolution. The active site contained electron density consistent with an intact molecule of **4** bound (Fig. 6a), which again displayed a $^2H_3$ half chair conformation, and identical interactions to those described for the complex with **3a**.

To obtain structural information on the interactions made between **4** and wild-type *Tm*GalA, crystals were grown and then soaked with the inhibitor for various periods of time. Following a 1 h soak with **4**, the structure, solved using data to 1.42 Å resolution, revealed electron density in the active site of *Tm*GalA consistent with the hydrolyzed carbasugar **6** (Fig. 6b) and active site interactions that were identical to those described above for the complex with **5**. Thus, within this timeframe **4** had been hydrolyzed with retention of stereochemistry. However, crystals that were soaked for a much shorter period of time (~2 min) with **4** led to a structure, solved using data to 1.77 Å resolution, where the electron density showed the covalent adduct of the carbasugar with the nucleophile Asp327 of *Tm*GalA (Fig. 6c). Of note, in these three structures (Fig. 6) there are no direct interactions between the ligand and any water molecule.

**Structural insights from QM/MM simulations**. Structures of the key stable states in the catalytic cycle for covalent inhibition of

*Tm*GalA by carbasugar **4** and the subsequent hydrolysis step were localized and optimized by means of hybrid QM/MM calculations (see Methods section). Snapshots of the active site corresponding to the Michaelis complex **E:I**, covalent intermediate **E-I**, and the Michaelis product complex **E:P**, obtained at density functional theory (DFT)/MM level are shown in Fig. 7. Following optimization of the enzyme-bound species on the reaction pathway, the conformations of the six-membered ring were classified using the Cremer–Pople ring puckering coordinates[46] and the corresponding angles used in Hill and Reilly method[47] (Supplementary Fig. 36 and Supplementary Table 4).

## Discussion

The minimal kinetic scheme for covalent inhibition of a GH by carbasugars is shown in Fig. 8. That is, the covalent labeling of *Tm*GalA by **4** involves rapid formation of the covalent intermediate ($k_3$), from the first-formed Michaelis complex (**E:I**), that is slowly hydrolyzed ($k_5$) to regenerate active enzyme (Fig. 8a). The appropriate equations for the full catalytic cycle involving enzyme-catalyzed turnover of **4** are given in Fig. 8b. Moreover, when the glycosylated intermediate accumulates ($k_3 \gg k_5$) the kinetic expressions for the time-dependent loss of enzyme activity are given in Fig. 8c. Notably, the second-order rate constant for

**a**

$$E + I \underset{k_2}{\overset{k_1}{\rightleftharpoons}} E{:}I \xrightarrow{k_3} EI \xrightarrow{k_5} E$$

**b**

$$k_{cat} = \frac{k_3 k_5}{k_3 + k_5}, \quad K_m = \frac{k_2 k_5}{k_1 k_3 + k_1 k_5} \quad \text{and} \quad \frac{k_{cat}}{K_m} = \frac{k_1 k_3}{k_2}$$

**c**

$$k_{inact} = k_3, \quad K_i = \frac{k_2}{k_1} \quad \text{and} \quad \frac{k_{inact}}{K_i} = \frac{k_1 k_3}{k_2}$$

**Fig. 8** Scheme for the covalent inhibition of α-glycosidases by cyclohexene carbasugar analogs. **a** Mechanistic scheme showing individual rate constants. **b** Kinetic expressions for hydrolysis of inhibitor. **c** Kinetic expressions for the covalent labeling stopped-flow experiments

the catalyzed turnover of substrate ($k_{cat}/K_m$) is identical to the corresponding rate constants for time-dependent inactivation ($k_{inact}/K_i$).

Considering that the two calculated second-order rate constants $k_{cat}/K_m$ and $k_{inact}/K_i$ are similar (the value for $k_{cat}/K_m$ assumes that the enzyme is 100% active, Table 1) and that no curvature is discernible in Supplementary Fig. 3b, we conclude that $k_{cat} \approx k_5$ for turnover of **4** and likely for **3b** as well. Thus the half-lives of the two covalent intermediates formed during the reactions of **3b** and **4** with *Tm*GalA are approximately 7 s and 20 min, respectively. Here the introduction of the 2-fluoro group retards hydrolysis of the covalent intermediate by a factor of about 150, whereas the second-order rate constant ($k_{cat}/K_m$) for formation of the covalent intermediate for the reaction of **3b** is 400-fold larger than that for **4** (Table 1). That is, the fluorine substitution shows an approximate two-fold larger rate reducing effect on covalent labeling of the enzyme than on dealkylation of the resulting covalent intermediate. In contrast to the established utility of activated 2-deoxy-2-fluoro-β-D-glycosides as covalent inhibitors of β-glycosidases[26], the corresponding 2,4-dinitrophenyl 2-deoxy-2-fluorogalactoside (**7**) is rapidly hydrolyzed by *Tm*GalA through a reaction involving rate-determining glycosylation (Table 1, $k_3 < k_5$). Conversely, we found that cyclophellitol **8** is a slow time-dependent inactivator of *Tm*GalA (Table 1, Supplementary Figs. 6 and 7) with no reactivation of the covalent enzyme intermediate over the course of 30 h ($k_5 \approx 0$).

Interestingly, these three distinct types of mechanism-based inhibitors exhibit completely different reactivities. First, our carbasugar analogs (**3a**, **3b**, and **4**) are covalent inhibitors that rapidly alkylate *Tm*GalA to give a covalently bound intermediate that is slowly hydrolyzed to regenerate active enzyme. That is, our compounds have the potential to show temporal effects of GH activity and offer unique advantages as probe molecules in cells. Second, cyclophellitol analog **8** is an inactivator of *Tm*GalA and thus other derivatives of this structural motif, such as the aziridines pioneered by Overkleeft[44], can be used as chemical biology probes for enzyme localization and quantification, though not as modulators of enzyme activity. Lastly, the 2-deoxy-2-fluoroglycoside **7** proved to be a substrate for *Tm*GalA and thus has little utility as a probe molecule.

The X-ray diffraction experiments provide an insight into the enzyme-bound species and afford three structures that define the reaction pathway for the two inversion events that occur at the pseudo-anomeric center. These data enable us to analyze the conformational changes that take place during the catalytic cycle for the reaction of our cyclohexene covalent inhibitors. Specifically, the formation of a covalent linkage between the pseudo-anomeric carbon of **4** and the nucleophile of *Tm*GalA (Asp327) confirms that this reaction occurs with inversion of

stereochemistry. Notably, the carbasugar ring binds in a flattened half-chair ($^2H_3$) conformation. An overlay of this structure with *Tm*GalA in complex with intact **4** reveals that formation of the covalent bond between the pseudo-anomeric carbon (C1) and Asp327 necessitates significant movement of C1 (2.04 Å), C2, and C6 (the carbon atom in the position of the endocyclic oxygen) and to a lesser extent C3, C4, and C5 (Fig. 9a). We note that all interactions with active site residues for the carbasugar remain unchanged when compared to the complex with **6**.

Hydrolysis of the covalent intermediate gives the hydrolyzed carbasugar **6**, which binds in a similar position and with the same conformation ($^2H_3$) as that observed for the carbasugar ring of the Michaelis complex with **4** (Fig. 9b). Specifically, a significant motion of C1, C2, and C6 returns the cyclohexene ring to close to its original position in the first Michaelis complex (Fig. 9c). In the complex with **6**, the pseudo-anomeric carbon is displaced by 0.39 Å compared to its position in the complex of the D387A mutant with **4**.

The DFT/MM calculations show that the carbasugar ring of **4** in the initial Michaelis complex (Fig. 7a) has a $^2H_3$ half chair conformation, in agreement with the X-ray structure. This $^2H_3$ half chair is slightly deformed towards a $^4H_3$–$^1S_3$ conformation in the fragment of **4** covalently bound to the nucleophile Asp327 in the covalent intermediate (Fig. 7b). Of note, the DFT/MM optimized structures and the conformations deduced from the X-ray structure exhibit very similar puckering indices. Finally, the DFT/MM optimized structure of the Michaelis product complex (Fig. 7c) display a $^2H_3$ half chair conformation while the X-ray structure is classified as lying between a $^2H_3$ half chair and a $^2E$ conformation. We extended our comparison between the QM/MM and X-ray structures to the analysis of the interatomic distances between the sugar and the residues of the active site (Supplementary Tables 6 and 7). Again, the agreement between experiments and calculations is remarkable with both techniques basically describing the same pattern of interactions. In addition, we computed the energies for the interactions that occur within the enzymatic active site between the inhibitor and the protein, an analysis that is based on the QM/MM interaction energies as shown in Supplementary Figs. 37 and 38. We found that interactions with Trp190, Asp220, and Lys325 stabilize all species along the reaction pathway. Also, Asp327 mediates a long-range interaction between Trp65 and Trp257 that stabilizes the leaving group fragment in the Michaelis complex **E:I**, an interaction that is likely perturbed in the *Tm*GalA D327A mutant used in the crystallographic experiments. The total sugar–protein interaction energies computed from the QM/MM and the X-ray structures also confirm the agreement between the results obtained from these two complementary techniques (Supplementary Fig. 38).

In summary, we synthesized two cyclohexene carbasugars that mimic both galactose and 2-deoxy-2-fluorogalactose and have determined their kinetic parameters for turnover and inhibition with *Tm*GalA. We show that our compounds are covalent inhibitors but importantly not covalent inactivators. As a result, our compounds inhibit GHs in a temporal fashion and enzyme activity returns when the covalent–enzyme intermediate is hydrolyzed. This is not the case for the cyclophellitol aziridine, cyclosulfate inactivators, and other inactivators that include the conduritol epoxides and most of the 2-deoxy-2-fluoro-β-glycosides. Therefore, our covalent inhibitors are functionally distinct to the listed covalent inactivators and in principal allow researchers to monitor cellular responses as enzyme activity decreases and then subsequently returns to baseline levels. In addition, we have provided structural insights into the conformations of the inhibitors at each step through the catalytic cycle derived from X-ray diffraction analysis as well as from

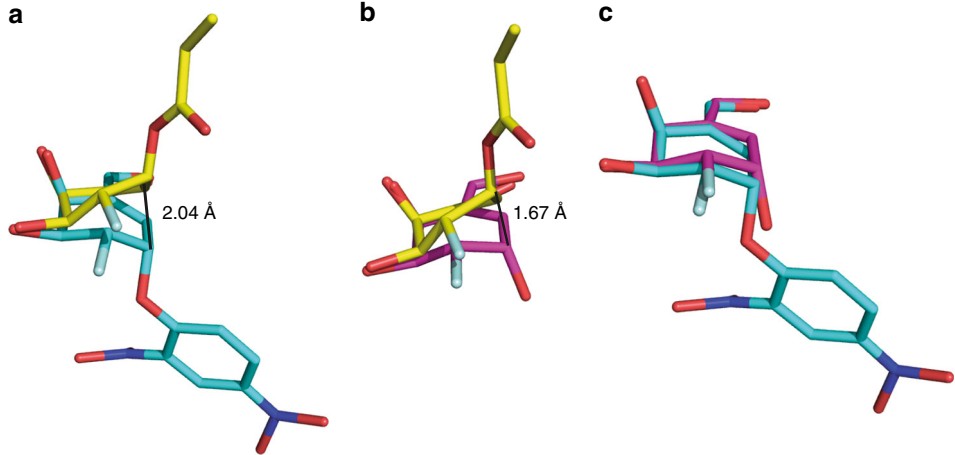

**Fig. 9** Superposition of *Tm*GalA structures. **a** Structure of *Tm*GalA mutant D387A in complex with intact **4** (cyan) and of wild-type *Tm*GalA in complex with 2-deoxy-2-fluorocarbagalactose fragment of **4** covalently bound to the nucleophile Asp327 (yellow). **b** Structure of wild-type *Tm*GalA in complex with hydrolyzed inhibitor **6** (pink) and of wild-type *Tm*GalA in complex with 2-deoxy-2-fluorocarbagalactose fragment of **4** covalently bound to the nucleophile Asp327 (yellow). **c** Structure of *Tm*GalA mutant D387A in complex with intact **4** (cyan) and of wild-type *Tm*GalA in complex with hydrolyzed inhibitor **6** (pink)

theoretical QM/MM calculations. Together these findings provide a glimpse into the subtleties of GH catalysis, which will aid the exploration of GH mechanisms and direct development of carbocyclic covalent inhibitors and inactivators of GHs.

## Methods

**Protein expression and purification**. A mutation, D387A, was introduced into the plasmid encoding *Tm*GalA using the QuikChange Site-directed Mutagenesis Kit (Stratagene, La Jolla, CA) using primers ForD387A, 5′-GATGAGGATAGGACC TGCTACTGCGCCGTTCTGGG-3′ and RevD387A, 5′-CCCAGAACGGCGCA GTAGCAGGTCCTATCCTCATC-3′. The plasmid encoding either α-galactosidase *Tm*GalA or *Tm*GalA with the D387A mutation was transformed into *Escherichia coli* BL-21(DE3) cells. *Tm*GalA and the D387A mutant were expressed recombinantly by growing cultures at 37 °C in Luria-Bertani broth containing kanamycin (50 μg mL$^{-1}$), until an optical density at 600 nm of approximately 0.6 absorbance units was reached. Overexpression was induced by the addition of 0.5 mM isopropyl β-D-1-thiogalactopyranoside, and cells were cultured for a further 4 h at 37 °C. Cells were harvested by centrifugation, re-suspended in phosphate-buffered saline (PBS), pH 7.4, 20 mM imidazole, and lysed using a cell disruptor at 30 kpsi. *Tm*GalA was applied to a nickel affinity chromatography column (5 mL HisTrap FF, GE Healthcare), washed with 10 column volumes of PBS, pH 7.4, and 50 mM imidazole, and eluted with 5 column volumes of PBS, pH 7.4, and 250 mM imidazole. *Tm*GalA was buffer exchanged into 20 mM HEPES, pH 7.4, 150 mM NaCl (HiPrep 26/10 desalting column, GE Healthcare) and then applied to a size-exclusion column (Superdex 200 16/60, GE Healthcare) for further purification. *Tm*GalA was judged to be >95% pure by sodium dodecyl sulfate-polyacrylamide gel electrophoresis.

**Enzyme kinetics**. Michaelis–Menten kinetic parameters for the hydrolysis of the cyclohexene carbasugar mimics of galactose (**3**), 2-deoxy-2-fluorogalactose (**4**) and the substrate 2,4-dinitrophenyl 2-deoxy-2-fluorogalactoside (**7**) were determined from a minimum of six initial rate measurements using a concentration range of at least $K_m/4$ to $4 \times K_m$. The progress of each reaction was monitored continuously for 5 min at 400 nm using a Cary 300 UV-vis spectrometer equipped with a temperature controller. Each 500 μL reaction mixture was prepared by addition of the appropriate volume of buffer (50 mM HEPES buffer, pH 7.4, $T = 37$ °C), substrate and enzyme. The rate versus substrate concentration data were fit to a Michaelis–Menten equation using a standard nonlinear least-squares computer program (Prism 7.0).

All covalent inhibition experiments with **4** were performed at 37 °C in 50 mM HEPES buffer, pH 7.4 using an Applied Photophysics SX18 stopped-flow spectrophotometer, equipped with an external temperature controller. The stopped-flow spectrometer was used in the sequential double mixing mode, in which rapidly mixed enzyme and inhibitor **4** were incubated for various time intervals prior to rapid mixing of the enzyme/inhibitor solution with a buffered solution of 4-nitrophenyl α-D-galactopyranoside. The remaining enzyme activity was monitored at a wavelength of 400 nm. Pseudo first-order rate constants for loss of enzyme activity ($k_{obs}$) at each inhibitor concentration were calculated by fitting the absorbance versus time data to a standard first-order rate equation using a nonlinear least squares routine in Prism 7.0. The kinetic parameters for inactivation by cyclophellitol **8** were determined using a classical dilution assay that

involved preincubation of the enzyme with varying concentrations of inhibitor at 37 °C in 50 mM HEPES buffer, pH 7.4 containing bovine serum albumin (BSA; 1 mg mL$^{-1}$). The remaining enzyme activity was measured periodically by removing an aliquot (10 μL) and adding it to a pre-equilibrated solution (37 °C) containing 4-nitrophenyl α-D-galactopyranoside (250 μM) in HEPES buffer (50 mM, pH = 7.4, [BSA] = 1 mg mL$^{-1}$). The first-order rate constants for inactivation ($k_{obs}$) were determined by fitting the absorbance versus time data to a standard first-order rate equation. The first- and second-order rate constants ($k_{inact}$ and $k_{inact}/K_i$) for inactivation were calculated by fitting the rate constant data versus the inactivator concentration to a standard Michaelis–Menten equation (Prism 7.0).

**Crystallization**. *Tm*GalA (10 mg mL$^{-1}$) was crystallized from 0.2 M MgSO$_4$ and 20% (w/v) poly(ethylene glycol) (PEG) 3350, and crystals were soaked in 30% (w/v) PEG 3350 containing 1 mM inhibitor **3a** overnight or in 0.2 M MgSO$_4$ and 30% (w/v) PEG 3350 containing a minute amount of inhibitor **4** added as powder for either 2 min or 1 h. The *Tm*GalA mutant D387A (10 mg mL$^{-1}$) was crystallized from 0.2 M MgSO$_4$ and 20% (w/v) PEG 3350, and crystals were soaked in 30% (w/v) PEG 3350 containing 2 mM inhibitor **3a** for 2 h. The *Tm*GalA mutant D387A (10 mg mL$^{-1}$) was incubated with ~1 mM inhibitor **4** and co-crystallized from 0.2 M MgSO$_4$ and 20% (w/v) PEG 3350. All crystals were cryo-protected in 0.2 M MgSO$_4$ and 30% (w/v) PEG 3350 prior to vitrification in liquid nitrogen.

**Data collection and processing**. X-ray diffraction data were collected at Diamond Light Source (DLS) (*Tm*GalA mutant D387A in complex with **3a**, beamline I04-1; *Tm*GalA in complex with **5**, beamline I04-1; *Tm*GalA in covalent complex with fragment of **4**, beamline I03) or the European Synchrotron Radiation Facility (*Tm*GalA mutant D387A in complex with **4**, beamline ID29; *Tm*GalA in complex with **6**, beamline ID29). Data processing and refinement statistics are listed in Supplementary Tables 1 and 2. Diffraction data were processed using the Xia2[48] pipeline to run DIALS or XDS[49] with Aimless[50] from the CCP4 suite[51]. Molecular replacement was performed using MOLREP[52] or Phaser[53] with Protein Data Bank (PDB) entry 5M0X as the search model. Refinement was performed using REFMAC5[54] and manual model building was done using Coot[55]. Structures were validated using PDB_REDO[56]. Models for the inhibitors were built in JSME[57] and the library was generated with PRODRG[58].

**Starting structure for theoretical simulations**. The starting structure for the computer simulations of the inhibition and hydrolysis of inhibitor **4** by *Tm*GalA) was the structure PDB code 5M12[23]. This structure corresponds to the structure of *Tm*GalA in complex with an intact cyclopropyl carbasugar. The structure of the cyclopropyl carbasugar was modified to correspond to inhibitor **4**. Missing atoms of Glu80 and Lys77 residues in X-ray structure were added using Accelrys Discovery Studio Visualizer v 4.5[59]. The charges and parameters for inhibitor **4**, obtained using the Antechamber software package along with a general AMBER force field (GAFF)[60], are listed in Supplementary Table 3. Hydrogen atoms were added using the tLEAP[61] module of Amber Tools program. The protonation state of titratable amino acids at pH 7.4 was determined using p$K_a$ results calculated using PROPKA ver. 3.1[62] available on PDB2PQR server[63]. The results indicate that Asp 387 and Glu 459 are present in their protonated form. Additionally, Glu224 was protonated to allow more favorable hydrogen bonding between neighboring Asp221 and inhibitor **4**. Furthermore, His30 and His273 were protonated at the

δ-position, while the rest of histidine residues were protonated at the ε-position. Neutralization of the total charge of the system was made by addition of 17 sodium cations ($Na^+$) in the electrostatically most favorable positions. Next, the system was placed in orthorhombic box of TIP3P[64] water molecules with size of $89 \times 96 \times 79$ Å³. The geometries of the remaining water molecules were then optimized. The full system was formed by the protein (8453 atoms), the substrate (37 atoms in **E:I**, 23 atoms in **E-I** and 25 atoms in **E:P**), and 17,398 solvation water molecules (52,194 atoms).

**MM MD simulations**. After performing the A387D reverse mutation and setting up the model, NAMD software[65] with the AMBER force field[66] was used to run classical molecular dynamics (MD) simulations with the full solvated protein to equilibrate the system. In all, 10 ns of classical MD simulation (at temperature 310 K) was carried out in the NVT ensemble using the Langevin–Verlet algorithm to equilibrate the system at the **E:I** state, after heating from 0 to 310 K with 0.001 K temperature increment. Periodic boundary conditions using the particle mesh Ewald method were applied. In order to improve time of calculations, a cut-off for nonbonding interactions was applied using a smooth switching function with between 14.5 to 16 Å. The time dependence of root mean square deviation, temperature, total energy, and fluctuation of Cα positions during the MD simulation are plotted in Supplementary Figure 34, while a graphical representation of the *B*-factor values on the protein as well as its population analysis are reported in Supplementary Figure 35. The results indicate that the system can be considered equilibrated after 10 ns of the MD simulation. The structure of TmGalA appears to be quite rigid, with no dramatic fluctuations during MD simulations (the maximum value of *B*-factor was 7.4 Å²).

**QM/MM optimizations**. The last structure from the 10 ns MM MD simulation of **E:I** complex was used to run the QM/MM calculations using fDynamo library[67] (Supplementary Figs. 39 and 40). In order to reduce time of calculations, positions of atoms presented beyond 20 Å from the inhibitor **4** were fixed. The QM subset of atoms were described first by the semiempirical AM1 Hamiltonian and later with the M06-2×[68] hybrid functional and the 6-31+G(d,p) basis set, as described below. The protein and water molecules were described with the OPLS[69] and TIP3P[64] classical force fields, respectively. fDynamo library[67] combined with Gaussian 09[70] were used for all the QM/MM calculations. Structures of the corresponding stable states for the formation of the covalent intermediate between the inhibitor **4** and the protein and its hydrolysis, **E:I**, **E-I**, and **E:P**, were first optimized at AM1/MM level of theory. **E-I** and **E:P** were located from the previously equilibrated Michaelis complex, **E:I**, by modifying the interatomic distances $d(OD2^{Asp327}-C1)$, $d(C1-O^{Lg})$, $d(OD2^{Asp387}-H^{Asp387})$, and $d(H^{Asp387}-O^{Lg})$. Thus, once the covalent bond between Asp327 and C1 atom of the sugar ring of inhibitor **4** was formed (intermediate **E-I:Lg**), the leaving group, **Lg**, was removed from the system and the cavity was filled with 5 water molecules. Next, 500 ps of QM/MM MD was run where the position of all atoms beyond distance of 20 Å from the substrate was fixed, thus generating the **E-I** intermediate after equilibration. Finally, the hydrolysis of this covalent intermediate and generation of compound **6** in the active site, **E:P**, was obtained by controlling the distances $d(O^{WAT}-C1)$, $d(C1-O^{Asp327})$, $d(OD2^{Asp387}-H^{WAT})$ and $d(H^{WAT}-O^{Asp387})$. The structure was then fully optimized without constraints.

Both Asp327 and Asp387 together with full inhibitor **4** were included in the QM region to locate the covalent intermediate, **E-I**. The two aspartate residues, the remaining part of inhibitor and one water molecule were described at QM level in the generation of the hydrolyzed inhibitor in the cavity of the protein, **E:P**. To saturate the valence of the QM/MM frontier atoms, link atoms were placed in the frontier QM–MM bonds of the two aspartate residues, as depicted in Supplementary Figure 39. Finally, the three key structures (**E:I**, **E-I**, and **E:P**) were optimized at M06-2×/OPLS-AA/TIP3P level, using the 6-31+G(d,p) basis set for the treatment of the QM subset of atoms.

**QM/MM MD simulations**. After the structures of the stationary points, Michaelis complex (**E:I**), covalent intermediate (**E-I**), and products (**E:P**) were optimized, 100 ps AM1/MM MD simulations were done in order to obtain statistical representation of these structures. Subsequent analysis of geometries and interaction energies was done based on 1000 structures.

Puckering coordinates of the ring: Two different formalisms have been used to define the puckering of the ring in the three key states of the full conformational itinerary: the one based on the Cremer–Pople puckering coordinates[46] and the angles used in the Hill and Reilly method[47]. The results are reported in Supplementary Table 4 and in Supplementary Figure 36.

**Individual interaction energies by amino acid**. The contributions of individual amino acid residues to the inhibitor stabilization were computed (in kcal mol⁻¹) as an average of the total QM–MM interaction energy over 1000 structures generated along the AM1/MM MD simulations initiated from optimized structures for the different TmGalA structures: intact **4**; 2-deoxy-2-fluorocarbagalactose fragment of **4** covalently bound to the nucleophile Asp327; and inhibitor **6** (see Supplementary Table 8 and Supplementary Figs. 37 and 38). Owing to the fact that this magnitude is directly obtained from the QM–MM interaction term of the full QM/MM Hamiltonian, only

the atoms of compound **4** were included in the QM region (not Asp327 or Asp387) in the first case, **E:I** (Supplementary Fig 37A). For the **E-I** state (Supplementary Fig. 37B) and since the fragment of compound **4** was covalently bounded to the protein, only this fragment of the compound and the sidechain of Asp327 were included in the QM region. Finally, for the **E:P** state (Supplementary Fig. 37C), only the hydrolyzed inhibitor, compound **6**, was included in the QM region.

The analysis of the interactions energies in the **E-I** state as plotted in Supplementary Figure 37B has to be carried out with caution due to the fact that Asp327 is covalently bound to the fragment of **4**, and the interactions that appear in the figure can be established with this residue and not with the fragment of inhibitor **4**. This is the case of Trp65 and Arg383, the repulsive interaction with Asp425, or the overestimated stabilizing interaction established with the positively charged Lys325. Nevertheless, as observed, the remaining favorable interactions are in agreement with the geometrical analysis that can be deduced from the interatomic distances (Supplementary Tables 5–7). Finally, when comparing the interactions of **6** (Supplementary Fig. 37C) with those established in **5**, less intense interactions are measured but a similar pattern is detected.

## Data availability

The accession codes for the structures reported in this article are: 6GTA, 6GVD, 6GWF, 6GWG, and 6GX8. All other data that support the findings of this study are available from the corresponding authors upon reasonable request.

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

## Acknowledgements

Financial support from the Natural Sciences and Engineering Research Council (NSERC) of Canada (AJB Discovery Grants: 121348–2012 & 2017–04910) was received. T.M.G. and V.O. are funded by a Wellcome Trust Career Development Fellowship, R.P. by Wellcome Trust ISSF, M.M. by a NSERC CGSD, M.F.-D. by a NSERC CGS-MSFSS and a GlycoNet Research Exchange Program, and R.B. by NSERC Discovery Grant and by a MSFHR Career Investigator Award. V.M. and K.Ś. thank the Spanish Ministerio de Economía y Competitividad and FEDER funds (project CTQ2015-66223-C2) and a Juan de la Cierva – Incorporación (ref. IJCI-2016-27503) contract, respectively, and Universitat Jaume I (project UJI-B2017-31). We thank the staff at DLS and ESRF synchrotrons for assistance with data collection.

## Author contributions

R.B. conceived of the de novo synthetic route. W.R., J.D. and M.M. assisted in planning and execution of the synthesis of covalent inhibitors. W.R. synthesized the 2-deoxy-2-fluorogalactoside and the cyclophellitol analog. R.P. and V.O. performed single crystal X-ray diffraction experiments. R.P. and T.M.G. analyzed X-ray diffraction data. S.S.K.A. and O.A. performed the kinetic experiments, S.C. made compound **3a**. K.Ś. and V.M. designed the computational studies and analyzed the results. K.Ś. and M.F.-D. carried out the

calculations. A.J.B. designed the covalent inhibitors and the overall study. A.J.B., T.M.G., R.B. and V.M. contributed to writing the paper. W.R. assisted with editing the manuscript.

## Additional information

**Competing interests:** The authors declare no competing interests.

