## [Peer Review File · Nature Communications]

Reviewers' comments:

Reviewer #1 (Remarks to the Author):

In this manuscript, Bennet et al. report two novel, covalent glycosidase inhibitors based on a cyclohexenol scaffold, and their activity against an α -galactosidase from *Thermotoga maritima* (TmGalA). The manuscript covers three main aspects:

- (i) the chemical synthesis of the inhibitors;
- (ii) their covalent inhibition kinetics with TmGalA;
- (iii) the structural characterization of the two inhibitors with TmGalA, including the structure of the covalent adduct formed during the turnover of one of the inhibitors.

Taken together, the results from (ii) and (iii) provide a conformational itinerary for the inhibitors along the kinetic reaction pathway.

This is a rigorous study that has been carried out to a very high technical standard. I have no doubt that the results will be of interest in particular to the biochemistry, chemical biology and medicinal chemistry communities.

However, there is already ample precedent for covalent inhibitors and probes for glycosidases, many of which have been characterized kinetically and structurally. Extensive information is therefore already available about the reaction itineraries for various classes of covalent glycosidase inhibitors, such as cyclophellitol aziridine and cyclosulfate inhibitors. This area has been reviewed very recently e.g. by Breen et al. (January 2018) *Competitive and Covalent Inhibitors of Human Lysosomal Retaining Exoglycosidases*. In: eLS. John Wiley & Sons, Ltd: Chichester. DOI: 10.1002/9780470015902.a0027591, and in an authoritative review by Rempel and Withers in *Glycobiology* 2008, 18(8) 570-586. Neither review has been cited in the present manuscript.

I am not certain that the results in the present manuscript significantly expand this previous work (beyond the introduction of a new inhibitor scaffold). To justify publication in *Nature Communications*, the authors ought to do one of two things, in my opinion:

- (i) demonstrate more clearly what we learn about glycosidases from their kinetic and structural studies with the new inhibitors that we didn't know from studies with other glycosidase inhibitors;
- (ii) demonstrate that these cyclohexenol-based inhibitors themselves have advantages over existing covalent glycosidase inhibitors, such as cyclophellitols, e.g. for applications in chemical biology.

MINOR

- Fig 8: reverse the order in which structures 2 & 3 are shown, so that they appear in the sequence of the reaction

- The introduction is a little long-winded and can be condensed. It should also focus more specifically on covalent glycosidase inhibitors (see comments above and literature references, which should be included), rather than glycosidase inhibitors in general.

- The synthetic chemistry is technically demanding and includes a number of interesting observations. However, it involves relatively conventional chemistry, which in itself does probably not justify publication in *Nat Comms*. The synthesis of the inhibitors should therefore be presented more concisely. Large sections can be moved to the SI or published elsewhere.

- The new inhibitors are based on cyclohexenols previously developed by the same group, the main difference being that the new inhibitors have a "galacto" instead of a "gluco" configuration. The design rationale behind the "galacto" configuration should be made clearer.

Reviewer #2 (Remarks to the Author):

Bennet and colleagues report on a well-executed, in-depth and insightful study on the way a new class of mechanism-based glycosidase inactivators work. The work focuses on retaining alpha-galactosidases however the results obtained should be transferable to other retaining glycosidases, specifically those hypothesised or known to follow similar reaction itineraries. The work takes inspiration from previous work both by the Bennet group (a recent publication in Nature Communications) in which related cyclopropanated cyclohexane derivatives stand central, and a JACS 2017 publication from Danby and Withers, which studies the hydrolysis of allylic cyclohexenes as catalysed by glycosidases. While the latter are substrates, substitution of the 2-OH (hexose numbering) by fluorine yields effective glycosidase inactivators. This OH-for-fluorine substitution is another page from the Withers book and I would be interested to see how the new compounds compare to a similarly (in terms of leaving group at the anomeric carbon) substituted 2-deoxy-2-fluoro-galactopyranoside in inactivating the target enzyme, this especially since the deoxy-fluorosugars are in some cases rather weak glycosidase inhibitors because of the lack of OH-2. Flattening of the inhibitor by virtue of the double bond may compensate for this loss, however this flattening can also be achieved by creation of the alpha-galactop analogue of the natural product and mechanism-based retaining glucosidase inhibitor, cyclophellitol (this galactose-configured cyclophellitol has been synthesised in the laboratory of the undersigned, JACS 2014, though reaction coordinates/structural studies as shown here were not included). There are, in other words, already a number of known retaining alpha-galactosidase inactivators out there, and including these in the storyline and in part also in experimental comparison would strengthen (in my opinion) the work, which otherwise is of such a quality and general interest that publication in Nature Communications is (again, in my opinion) warranted.

Herman Overkleeft

Reviewer #3 (Remarks to the Author):

Ren et al. have analyzed the mode of action of covalently acting cyclohexyl inhibitors on the family of glycoside hydrolase 36 (GH36). The authors suggest conformational changes during the conversion of the surrogates at the catalytic active site. It looks like the main focus of the manuscript is put in the description of the sophisticated compound synthesis though kinetic and crystallographic characterization have been performed to elucidate the reaction trajectory of this class of ligands. The applied strategy is comprehensive, but there is a lack of first-principle quantum chemical and classical molecular simulations. These calculations are essential to obtain profound insights into the structure, energetics, and dynamics of GH36:ligand complexes and would significantly strengthen the drawn conclusions. Of disadvantage is that the authors solely describe the characterization of one class of GH36 inhibitors, but they do not give any conclusion how to design useful chemical tools for related hydrolases in the near future.

Major concerns:

1) Proper synthesis of carbohydrates is demanding and respected among chemists. However the elaborate synthesis report (7 pages of description in the main text) for the 2,4-dinitrophenyl ether 3b or the 2-fluorocarbasugar 4 will not address the broad interdisciplinary readership and therefore is too specialized. Some chapters read like a Material & Methods section (see for example p10 and p12). On the other hand, the statement "full discussion of the scope and utility of this transformation will be presented elsewhere (p11, last sentence)" is inappropriate in an original paper and should be documented in detail in the SI.

2) Structural and functional characterization was solely performed with GalA from the hypothermophilic organism *T. maritima*. Since GHs are already promising targets for the development of therapeutics (validated by the influenza therapeutic oseltamivir), it is requested to also characterize GH36 enzymes from human pathogens. Moreover, the authors did not mention why a TmGH36D387A mutant was used for the crystallographic studies. Kinetic parameters have to be determined for this mutant as well.

3) The carbohydrates are well defined in the respective electron density maps shown in figures 7 and 8. However, this presentation only depicts the quality of the electron density for the respective ligand, but lacks any further information on the mode of binding. No interactions with protein

residues are shown (except Fig. 7c) and none of the amino acids have been labelled. Taken together the message of the figure illustrations is poor.

4) Evaluations of the kinetic parameters including stopped-flow experiments might have been carried out properly, though an expert in these techniques has to be consulted for accurate evaluation.

5) The authors included snapshots of the reaction coordinate by performing TmGH36D387A crystal soaking between 5s and 8h. The subsequent structure determinations requires reprocessing of some datasets: in the high resolution shell, Rmerge has to be <0.6 and $I/\sigma I > 2.0$. In addition, the authors have to address how the crystallization conditions such as pH, high magnesium concentrations and 20% polyethylene glycol may affect the GH36 activity. Therefore, it is requested to also determine the kinetic parameters of GH36s in the applied crystallization conditions. There should be a note about the temperature factors in the active site residues and the ligand versus solvent and the remaining part of the protein.

6) It is essential to include advanced atomic simulations to calculate binding energies for experimentally achieved poses with corresponding MM and QM/MM refined structures, etc.. In the present form of the manuscript, it is invalid to claim that "having three structures, that define the reaction coordinate for the two inversion events that occur at the pseudo-anomeric center, enabled us to analyze the motions and conformational changes that take place during the catalytic cycle for the reaction of our cyclohexene covalent inhibitors".

Reviewer #1 (Remarks to the Author):

In this manuscript, Bennet et al. report two novel, covalent glycosidase inhibitors based on a cyclohexenol scaffold, and their activity against an α -galactosidase from *Thermotoga maritime* (TmGalA). The manuscript covers three main aspects:

- (i) the chemical synthesis of the inhibitors;
- (ii) their covalent inhibition kinetics with TmGalA;
- (iii) the structural characterization of the two inhibitors with TmGalA, including the structure of the covalent adduct formed during the turnover of one of the inhibitors.

Taken together, the results from (ii) and (iii) provide a conformational itinerary for the inhibitors along the kinetic reaction pathway.

This is a rigorous study that has been carried out to a very high technical standard. I have no doubt that the results will be of interest in particular to the biochemistry, chemical biology and medicinal chemistry communities.

Comment 1: However, there is already ample precedent for covalent inhibitors and probes for glycosidases, many of which have been characterized kinetically and structurally. Extensive information is therefore already available about the reaction itineraries for various classes of covalent glycosidase inhibitors, such as cyclophellitol aziridine and cyclosulfate inhibitors. This area has been reviewed very recently e.g. by Breen et al. (January 2018) *Competitive and Covalent Inhibitors of Human Lysosomal Retaining Exoglucosidases*. In: eLS. John Wiley & Sons, Ltd: Chichester. DOI: 10.1002/9780470015902.a0027591, and in an authoritative review by Rempel and Withers in *Glycobiology* 2008, 18(8) 570-586. Neither review has been cited in the present manuscript.

Response: We show that our compounds are covalent inhibitors, but importantly not covalent inactivators. As a result, our compounds inhibit glycoside hydrolases in a temporal fashion and that enzyme activity returns when the covalent-enzyme intermediate is hydrolyzed. This is not the case for the cyclophellitol aziridine, cyclosulfate inactivators, and other inactivators that include the conduritol epoxides and most of the 2-deoxy-2-fluoro- β -glycosides. Therefore, our covalent inhibitors are functionally distinct to the listed covalent inactivators and in principal allow researchers to monitor cellular responses as enzyme activity decreases and then subsequently returns to baseline levels.

We have included the above two references in our revised manuscript.

Comment 2: I am not certain that the results in the present manuscript significantly expand this previous work (beyond the introduction of a new inhibitor scaffold). To justify publication in *Nature Communications*, the authors ought to do one of two things, in my opinion:

Suggestion 1: demonstrate more clearly what we learn about glycosidases from their kinetic and structural studies with the new inhibitors that we didn't know from studies with other glycosidase inhibitors;

Response: We have measured the kinetic parameters for two other families of glycoside inhibitors, the 2-deoxy-2-fluoroglycosides introduced by Withers and co-workers and the cyclophellitol type natural products. Also, see our response to comments 1 and 2 from reviewer #2 (Overkleeft) given below.

Suggestion 2: demonstrate that these cyclohexenol-based inhibitors themselves have advantages over existing covalent glycosidase inhibitors, such as cyclophellitols, e.g. for applications in chemical biology.

Response: We have pointed out the potential advantages for our covalent inhibitors in the revised manuscript. That is, our inhibitors are covalent inhibitors and thus complementary to covalent inactivators such as the cyclophellitols and their aziridine analogues.

MINOR

Comment 1: Fig 8: reverse the order in which structures 2 & 3 are shown, so that they appear in the sequence of the reaction

Response: We have redrawn this figure (Figure 6 in the revised manuscript) as suggested by this reviewer.

Comment 2: The introduction is a little long-winded and can be condensed. It should also focus more specifically on covalent glycosidase inhibitors (see comments above and literature references, which should be included), rather than glycosidase inhibitors in general.

Response: We have shortened the introduction as requested.

Comment 3: The synthetic chemistry is technically demanding and includes a number of interesting observations. However, it involves relatively conventional chemistry, which in itself does probably not justify publication in Nat Comms. The synthesis of the inhibitors should therefore be presented more concisely. Large sections can be moved to the SI or published elsewhere.

Response: To satisfy the comments of Reviewer No. 1, we have removed the retrosynthetic analysis (includes removal of Fig. 4 and renumbering of subsequent figures) and have shortened the discussion of compound synthesis. Considering that the synthetic strategy used to access the carbasugars relies on methods that have never been applied to the synthesis of a cyclohexenol analogue, we feel that the full discussion of this work is appropriate.

Comment 4: The new inhibitors are based on cyclohexenols previously developed by the same group, the main difference being that the new inhibitors have a “galacto” instead of a “gluco” configuration. The design rationale behind the “galacto” configuration should be made clearer.

Response: We have included the following statement in the revised manuscript: "*As a result, we hypothesized that a galacto-configured analogue would covalently label a GH36 α -galactosidase, a member of GH clan-D, that has as its main structural element a $(\beta/\alpha)_8$ fold, which is the same protein fold as found for family GH13*"; that is, the previous enzyme used by us is a glucosidase that is in family GH13 and the current enzyme is in family GH36, while the human enzyme studied by Overkleeft (see reviewer #2) is in the same clan as GH36.

Reviewer #2 (Remarks to the Author):

Bennet and colleagues report on a well-executed, in-depth and insightful study on the way a new class of mechanism-based glycosidase inactivators work. The work focuses on retaining alpha-galactosidases however the results obtained should be transferable to other retaining glycosidases, specifically those hypothesised or known to follow similar reaction itineraries. The work takes inspiration from previous work both by the Bennet group (a recent publication in Nature Communications) in which related cyclopropanated cyclohexane derivatives stand central, and a JACS 2017 publication from Danby and Withers, which studies the hydrolysis of allylic

cyclohexenes as catalysed by glycosidases. While the latter are substrates, substitution of the 2-OH (hexose numbering) by fluorine yields effective glycosidase inactivators.

Comment 1: This OH-for-fluorine substitution is another page from the Withers book and I would be interested to see how the new compounds compare to a similarly (in terms of leaving group at the anomeric carbon) substituted 2-deoxy-2-fluoro-galactopyranoside in inactivating the target enzyme, this especially since the deoxy-fluorosugars are in some cases rather weak glycosidase inhibitors because of the lack of OH-2.

Response: We have made 2,4-dinitrophenyl 2-deoxy-2-fluoro- α -D-galactopyranoside (details are in the revised supporting information) and measured its hydrolysis with the GH36 enzyme (*TmGalA*) used in this study. We note that this compound (2,4-dinitrophenyl 2-deoxy-2-fluoro- α -D-galactopyranoside) is not a covalent inhibitor of the GH36 enzyme but rather a poor substrate. That is, deglycosylation (k_5) is faster than glycosylation (k_3) so in this case the covalent intermediate does not accumulate.

Comment 2: Flattening of the inhibitor by virtue of the double bond may compensate for this loss, however this flattening can also be achieved by creation of the alpha-galactop analogue of the natural product and mechanism-based retaining glucosidase inhibitor, cyclophellitol (this galactose-configured cyclophellitol has been synthesised in the laboratory of the undersigned, JACS 2014, though reaction coordinates/structural studies as shown here were not included).

Response: We have made the cyclophellitol analogue (compound number **8** in the revised manuscript) as suggested and we have measured the kinetics for inactivation of our GH36 enzyme (*TmGalA*) by **8** and note that it is an inactivator of *TmGalA* (kinetic data for **7** and **8** have been added to Table 1)

Summary from Reviewer #2

There are, in other words, already a number of known retaining alpha-galactosidase inactivators out there, and including these in the storyline and in part also in experimental comparison would strengthen (in my opinion) the work, which otherwise is of such a quality and general interest that publication in Nature Communications is (again, in my opinion) warranted.

Herman Overkleeft

Reviewer #3 (Remarks to the Author):

Ren et al. have analyzed the mode of action of covalently acting cyclohexyl inhibitors on the family of glycoside hydrolase 36 (GH36). The authors suggest conformational changes during the conversion of the surrogates at the catalytic active site. It looks like the main focus of the manuscript is put in the description of the sophisticated compound synthesis though kinetic and crystallographic characterization have been performed to elucidate the reaction trajectory of this class of ligands. The applied strategy is comprehensive, but there is a lack of first-principle quantum chemical and classical molecular simulations. These calculations are essential to obtain profound insights into the structure, energetics, and dynamics of GH36:ligand complexes and would significantly strengthen the drawn conclusions. Of disadvantage is that the authors solely describe the characterization of one class of GH36 inhibitors, but they do not give any conclusion how to design useful chemical tools for related hydrolases in the near future.

General response: We have shown that the inhibitory properties of these cyclohexenyl-based covalent inhibitors can be tuned by changing a single hydroxyl group in the molecule. We believe that the data contained in this manuscript will stimulate other researchers to design

covalent inhibitors based on our compounds' ability to affect glycosidase activity in a temporal fashion (also see response to Reviewer #1, comment #1).

Concern 1: Proper synthesis of carbohydrates is demanding and respected among chemists. However the elaborate synthesis report (7 pages of description in the main text) for the 2,4-dinitrophenyl ether 3b or the 2-fluorocarbasugar 4 will not address the broad interdisciplinary readership and therefore is too specialized. Some chapters read like a Material & Methods section (see for example p10 and p12). On the other hand, the statement “full discussion of the scope and utility of this transformation will be presented elsewhere (p11, last sentence)” is inappropriate in an original paper and should be documented in detail in the SI.

Response: See response above to minor comment #3 from Reviewer No. 1, we have condensed this portion of the manuscript. We have also removed the last sentence on page 11.

Concern 2: Structural and functional characterization was solely performed with GalA from the hypothermophilic organism *T. maritima*. Since GHs are already promising targets for the development of therapeutics (validated by the influenza therapeutic oseltamivir), it is requested to also characterize GH36 enzymes from human pathogens. Moreover, the authors did not mention why a TmGH36D387A mutant was used for the crystallographic studies. Kinetic parameters have to be determined for this mutant as well.

Response: The aim of this paper was to determine the conformations of the intact covalent inhibitor, the covalent intermediate, and the hydrolyzed product, so as to be able to facilitate the design of better inhibitors. We chose the GH36 enzyme because in our hands we have been able to solve structures to high resolution (in the current paper the five structures have a resolution of 1.22, 1.42, 1.72, 1.77 and 2.20 Å) while to the best of our knowledge the highest resolution structure of the human GH27 α -galactosidase is 1.90 Å (Guce et al. Catalytic mechanism of human α -galactosidase, *J. Biol. Chem.* **2010**, 285, 3625–3632). With such high resolution data, we can be confident in the interpretations of sugar conformations in order to dissect the conformational itinerary. In addition, we feel confident in the use of the GH36 enzyme as a model for the human GH27 enzyme as these two glycoside hydrolase families are in Clan-D and enzymes in this clan have been shown to have common mechanistic features (see Comfort et al. Biochemical analysis of *Thermotoga maritima* GH36 α -galactosidase (TmGalA) confirms the mechanistic commonality of Clan GH-D glycoside hydrolases, *Biochemistry* **2007**, 46, 3319–3330).

We determined the kinetic parameters for the D387A mutant (Table 1). Specifically, the ratio of k_{cat} and k_{cat}/K_m values for wild-type to D387A mutant for the hydrolysis of 4-nitrophenyl α -D-galactopyranoside are ~900 and 3,200, respectively (revised manuscript).

Given the kinetic parameters measured by others (Comfort et al.) and during this work, which shows the acid/base mutant was catalytically less efficient compared to wild type enzyme than the nucleophile mutant, we decided to try this first for crystallography studies. We have added a short explanation for the use of the D387A mutant enzyme into the revised manuscript (along with the appropriate kinetic parameter in Table 1), where we state "*In order to obtain a complex with intact 3a, we employed a D387A mutant of TmGalA in which the acid/base aspartic acid is mutated to alanine, and it displays an approximate 3,100-fold reduction in its second order rate constant (k_{cat}/K_m) relative to wild-type for the hydrolysis of 4-nitrophenyl α -D-galactopyranoside (Table 1)*".

Concern 3: The carbohydrates are well defined in the respective electron density maps shown in figures 7 and 8. However, this presentation only depicts the quality of the electron density for the

respective ligand, but lacks any further information on the mode of binding. No interactions with protein residues are shown (except Fig. 7c) and none of the amino acids have been labelled. Taken together the message of the figure illustrations is poor.

Response: We have added residue labels to Figures 5 and 6 (which were figures 7 and 8 in the original submission), and included a new panel in Figure 5 which explicitly shows the hydrogen bond interactions in the active site.

Concern 4: Evaluations of the kinetic parameters including stopped-flow experiments might have been carried out properly, though an expert in these techniques has to be consulted for accurate evaluation.

Response: My group (AJB) has published several papers on stopped-flow kinetics including examples using the sequential mixing accessory. For example, we measured the fast kinetics ($k_{\text{inact}}/K_i = (3.9 \pm 0.8) \times 10^6 \text{ M}^{-1} \text{ s}^{-1}$) for inactivation of a sialidase: "Inhibitory Efficiencies for Mechanism-Based Inactivators of Sialidases" *Can. J. Chem.* **2015**, *93*, 1207–1213.

Concern 5: The authors included snapshots of the reaction coordinate by performing TmGH36D387A crystal soaking between 5s and 8h. The subsequent structure determinations requires reprocessing of some datasets: in the high resolution shell, Rmerge has to be <0.6 and I/sigmaI > 2.0. In addition, the authors have to address how the crystallization conditions such as pH, high magnesium concentrations and 20% polyethylene glycol may affect the GH36 activity. Therefore, it is requested to also determine the kinetic parameters of GH36s in the applied crystallization conditions. There should be a note about the temperature factors in the active site residues and the ligand versus solvent and the remaining part of the protein.

Response: We have measured kinetic parameters under conditions where the enzyme displays high activity. We have not made any attempt to measure activity in 20% polyethylene glycol for the following reasons: i) crystallization conditions are chosen in order that the enzyme crystallizes and so the amount of enzyme in solution may change with time, ii) in a medium of high viscosity reaction kinetics can become dominated by diffusional effects that are not seen under normal biological conditions.

Traditionally the measures for data quality and resolution cut-off were Rmerge and I/sigmaI, as suggested by the reviewer. However, over recent years, the pitfalls of using these terms have been highlighted, and it is now considered most acceptable in the protein crystallography field that a CC1/2 value of greater than 0.5 is used to determine the resolution cut-off. This is discussed further in articles that include Evans & Murshudov, *Acta Crystallogr D Biol Crystallogr.*, 2013, *69*, 1204 and Karplus & Diederichs, *Curr Opin Struct Biol.*, 2015, *34*, 60. All of our datasets have a CC1/2 value equal to or greater than 0.5.

Concern 6: It is essential to include advanced atomic simulations to calculate binding energies for experimentally achieved poses with corresponding MM and QM/MM refined structures, etc.. In the present form of the manuscript, it is invalid to claim that "having three structures, that define the reaction coordinate for the two inversion events that occur at the pseudo-anomeric center, enabled us to analyze the motions and conformational changes that take place during the catalytic cycle for the reaction of our cyclohexene covalent inhibitors".

Response: Following the reviewer's suggestion, an extensive study of the covalent inhibition and the hydrolysis of compound 4 has been explored by advanced QM/MM methods. In particular, after running Molecular Dynamics (MD) simulations to equilibrate the system, the formation of the covalent intermediate and the subsequent hydrolysis step have been explored for inhibitor 4 by computing the corresponding QM/MM Potential Energy Surfaces (PES). The three key stable

conformations, (Michaelis complex, covalent intermediate and the Michaelis product complex, E:I, E-I and E:P) were located first at the AM1/MM level. Then a complete optimization of the system without any constraints (full *TmGalA* protein solvated in a box of $89 \times 96 \times 79 \text{ \AA}^3$ with the corresponding substrate in the active site; 52194 atoms in total) was carried out at DFT/MM level (using the hybrid DFT functional of Truhlar and co-workers, M06-2X, with the 6-31+G(d,p) basis set) for the three stable states. The optimized structures, as well as averaged structures derived from additional QM/MM MD simulations have been compared with the X-ray structures (inter-atomic distance between the compound and residues of the protein, conformation of the carbasugar ring by means of the Cremer-Pople puckering coordinates and the corresponding angles used in the Hill and Reilly method, etc). Moreover, the protein-substrate binding energies decomposed by amino acids have been estimated for the different states. A new section has been introduced in the text and a complete description of the computational details can be found in the revised Supporting Information. These results, together with analysis of charges of the key atoms of the inhibitor computed at DFT/MM level provide a robust support to the X-ray experimental studies. A section has been included in the manuscript and a detailed description of methods and supporting results are deposited in the Supporting Information.

REVIEWERS' COMMENTS:

Reviewer #1 (Remarks to the Author):

The revised version of the manuscript by Ren et al addresses the key points that have been raised by the reviewers. In particular, it now includes a direct comparison of new covalent inhibitors 3b and 4, and previously reported covalent inactivators 7 and 8, with regard to their inhibition kinetics.

Three different types of reactivities were observed for the different inhibitors classes, and these are discussed in the revised manuscript, including implications of the different reactivities for the use of these inhibitors as tool compounds or potential therapeutics (p11 ff; please correct typo "probe molecules").

In their rebuttal, the authors emphasise the difference between covalent inhibitors and inactivators:

"We show that our compounds are covalent inhibitors, but importantly not covalent inactivators. As a result, our compounds inhibit glycoside hydrolases in a temporal fashion and that enzyme activity returns when the covalent-enzyme intermediate is hydrolyzed. This is not the case for the cyclophellitol aziridine, cyclosulfate inactivators, and other inactivators that include the conduritol epoxides and most of the 2-deoxy-2-fluoro- β -glycosides. Therefore, our covalent inhibitors are functionally distinct to the listed covalent inactivators and in principal allow researchers to monitor cellular responses as enzyme activity decreases and then subsequently returns to baseline levels."

This is a very concise and pertinent analysis of the differences between the new and existing covalent inhibitors/inactivators, and I recommend including it in the manuscript, e.g. in the summary.

Given the importance of the conceptual difference between covalent "inhibitor" and "inactivator", as highlighted by the authors, it may be useful to also include a brief definition of both terms, and to check that this terminology is applied consistently and correctly throughout the manuscript, e.g. the first sentence in the abstract: "Mechanism-based glycoside hydrolase (GH) inactivators..." should probably read "inhibitors" in this context.

I also note that the authors have carried out an extensive study of the covalent inhibition and the hydrolysis of compound 4 by advanced QM/MM methods, to address concerns raised by reviewer 3. These new data add substantially to the manuscript.

I am confident that with these changes, the revised manuscript is eminently suitable for publication in Nature Communications.

Reviewer #3 (Remarks to the Author):

Ren et al. have undertaken some efforts to implement the referees' comments. In my opinion, the crystallographic data remains poorly processed. The depiction of electron density maps will most likely not address a broad readership (especially figures 5A-C, 6A-C and 9), but just illustrate that additional x-ray analysis have been performed. The comments concerning cc1/2 and refinement seem to follow a trend to report resolutions by exaggerating numbers. It is not important if the resolution is 1.9 or 1.6 angstrom, as both electron densities will show ligand and water molecules. Unfortunately, the latter have been neglected, even though solvent is essential for theoretical simulations as mentioned on page 24: "The protein and water molecules were described with the OPLS [63,64] and TIP3P [65] classical force fields, respectively. fDynamo library [66] combined with Gaussian 09 [67] were used for the QM/MM calculations."

Taken together, I am no expert in the field of glycosidases and this work would not attract much of my attention in its current presentation. However, it appears that

referee 1 and 2 (is it reasonable that he identified by name Overkleeft prior to acceptance of the submission(?)) are specialists on glycosidases. Thus, I suggest that their recommendations should be taken into closer account than mine.

RESPONSE TO REVIEWERS

Reviewer #1 (Remarks to the Author):

The revised version of the manuscript by Ren et al addresses the key points that have been raised by the reviewers. In particular, it now includes a direct comparison of new covalent inhibitors 3b and 4, and previously reported covalent inactivators 7 and 8, with regard to their inhibition kinetics.

Comment 1: Three different types of reactivities were observed for the different inhibitors classes, and these are discussed in the revised manuscript, including implications of the different reactivities for the use of these inhibitors as tool compounds or potential therapeutics (p11 ff; please correct typo “probe molecules”).

Response: We have corrected this typographical error.

Comment 2: In their rebuttal, the authors emphasise the difference between covalent inhibitors and inactivators: “We show that our compounds are covalent inhibitors, but importantly not covalent inactivators. As a result, our compounds inhibit glycoside hydrolases in a temporal fashion and that enzyme activity returns when the covalent-enzyme intermediate is hydrolyzed. This is not the case for the cyclophellitol aziridine, cyclosulfate inactivators, and other inactivators that include the conduritol epoxides and most of the 2-deoxy-2-fluoro- β -glycosides. Therefore, our covalent inhibitors are functionally distinct to the listed covalent inactivators and in principal allow researchers to monitor cellular responses as enzyme activity decreases and then subsequently returns to baseline levels.” This is a very concise and pertinent analysis of the differences between the new and existing covalent inhibitors/inactivators, and I recommend including it in the manuscript, e.g. in the summary.

Response: We have incorporated the above sentences as suggested by reviewer 1 into the last paragraph of the discussion.

Comment 3: Given the importance of the conceptual difference between covalent “inhibitor” and “inactivator”, as highlighted by the authors, it may be useful to also include a brief definition of both terms ...

Response: We have added the short sentence to the start of the results section as suggested by reviewer 1: "That is, covalent inhibitors result in reversible loss of enzyme activity, a process that is irreversible with covalent inactivators."

Comment 4: continued from **comment 3**, ... and to check that this terminology is applied consistently and correctly throughout the manuscript, e.g. the first sentence in the abstract: “Mechanism-based glycoside hydrolase (GH) inactivators...” should probably read “inhibitors” in this context.

Response: This change has been incorporated into the revised abstract and we have checked all uses of covalent inhibitor and inactivator throughout the manuscript and corrected where necessary.

Reviewer #3 (Remarks to the Author):

Comment 1: Ren et al. have undertaken some efforts to implement the referees' comments. In my opinion, the crystallographic data remains poorly processed. The depiction of electron density maps will most likely not address a broad readership (especially figures 5A-C, 6A-C and 9), but just illustrate that additional x-ray analysis have been performed. The comments concerning cc1/2 and refinement seem to follow a trend to report resolutions by exaggerating numbers. It is not important if the resolution is 1.9 or 1.6 angstrom, as both electron densities will show ligand and water molecules. Unfortunately, the latter have been neglected, even though solvent is essential for theoretical simulations as mentioned on page 24: "The protein and water molecules were described with the OPLS [63,64] and TIP3P [65] classical force fields, respectively. fDynamo library [66] combined with Gaussian 09 [67] were used for the QM/MM calculations."

Response: We have added to the legend for Figure 5 to emphasis the water molecule. We have also added two short sentences to emphasis that no water molecules interact with the carbasugar in the enzyme active site (except that specified explicitly in Fig. 5). We have added B-factors to the X-ray data tables in the appropriate Supplementary Tables.